# Unveiling Chain of Step Reasoning for Vision-Language Models with Fine-grained Rewards

**Honghao Chen**[1,2,3*] **Xingzhou Lou**[1,2*] **Xiaokun Feng**[1,2*] **Kaiqi Huang**[1,2†] **Xinlong Wang**[3†]

[1]Institute of Automation, Chinese Academy of Sciences
[2]School of Artificial Intelligence, University of Chinese Academy of Sciences
[3] Beijing Academy of Artificial Intelligence

## Abstract

Chain of thought reasoning has demonstrated remarkable success in large language models, yet its adaptation to vision-language reasoning remains an open challenge with unclear best practices. Existing attempts typically employ reasoning chains at a coarse-grained level, which struggles to perform fine-grained structured reasoning and, more importantly, are difficult to evaluate the reward and quality of intermediate reasoning. In this work, we delve into chain of step reasoning for vision-language models, enabling assessing reasoning step quality accurately and leading to effective reinforcement learning and inference-time scaling with fine-grained rewards. We present a simple, effective, and fully transparent framework, including the step-level reasoning data, process reward model (PRM), and reinforcement learning training. With the proposed approaches, our models set strong baselines with consistent improvements on challenging vision-language benchmarks. More importantly, we conduct a thorough empirical analysis and ablation study, unveiling the impact of each component and several intriguing properties of inference-time scaling. We believe this paper serves as a baseline for vision-language models and offers insights into more complex multimodal reasoning. Our dataset, PRM, and code at `https://github.com/baaivision/CoS`.

## 1 Introduction

The complex reasoning capabilities of large language models (LLMs) have recently garnered widespread attention [57, 8, 47, 68, 1]. By generating a detailed chain of thought (CoT) process before producing the final answer, LLMs enable effective test-time scaling and significantly enhancing the reasoning and logical capabilities. The integration of CoT reasoning with large-scale reinforcement learning [43] has further led to revolutionary advancements, exemplified by groundbreaking models such as OpenAI-o1 [17] and DeepSeek-R1 [12]. Inspired by the success of NLP, many studies focus on exploring complex reasoning methods in vision-language models (VLMs), achieving various advances [66, 58, 10, 55, 34, 38, 29]. Compared to LLMs, the presence of visual information in VLMs introduces greater complexity and plays a more critical role, emphasizing the necessity of leveraging visual cues for precise step-by-step reasoning and problem-solving.

Despite the advances, enabling VLMs to perform human-level reasoning remains a key challenge [61]. A fundamental difference is that humans typically engage in systematic and structured thinking [48], whereas the CoT outputs of current VLMs are constrained to coarse-grained reasoning, making it difficult to execute systematic and structured inference [58]. Specifically, current VLMs typically output a lengthy thought without a unified structure or clear step-by-step progression. This form

---

*Equal contribution. Work done during Honghao's internship at BAAI.
†Corresponding authors.

of reasoning tends to become disorganized or verbose and, more importantly, makes it difficult to evaluate the quality of reasoning. As a result, selecting high-quality reasoning trajectories for efficient training (e.g., reinforcement learning) or inference becomes challenging. It is natural to ask whether step-by-step fine-grained reasoning can facilitate better training or inference. However, answering this is non-trivial, as it involves two major challenges: **i)** defining step, i.e., how to decompose the overall reasoning chain into logically coherent and progressive steps; **ii)** evaluating step, i.e., how to provide fine-grained reward signals for step-wise reasoning to facilitate training and inference.

To address these challenges and answer the above question, we propose Chain-of-Step reasoning and introduce corresponding supervised fine-tuning (SFT) and reinforcement learning strategies to enhance VLMs' reasoning capabilities. We start by defining step: the entire reasoning chain is decomposed into consecutive steps, with each step consisting of three components: Name, Thought, and Reflection, separated by special tags. The Name component provides a concise summary of the step, the Thought component contains detailed reasoning specific to that step, and the Reflection component establishes connections between the current reasoning step, the visual information, and previous steps. Our design boasts three essential advantages: **i)** structuring reasoning into discrete steps to ensure clarity; **ii)** maintaining a concise and well-organized structure; **iii)** incorporating visual content along with reflective reasoning. We use GPT-4o [16] to construct ShareGPT-Step-300K, a dataset of 300K structured step-wise reasoning samples following the designed template. Our dataset encompasses a diverse range of tasks, utilizing 17 datasets that demand various reasoning skills, such as scientific reasoning [18, 31], mathematical reasoning [51, 64] and world knowledge [41].

Through SFT on ShareGPT-Step-300K, VLMs can output step-by-step reasoning chains. However, reinforcement learning is more crucial for enhancing the model's reasoning capabilities [66], and providing accurate rewards is of vital importance [34]. Therefore, we delve into the precise evaluation of reasoning steps. We collect process-annotated data using two mainstream methods: Monte Carlo estimation and LLM-as-Judge [67]. Specifically, we employ Math-Shepherd [52] and GPT-4o [16] to annotate the step-wise reasoning trajectories, which we then use to train a process reward model (PRM). Based on PRM, we can assess the quality of intermediate reasoning steps with high accuracy, thereby providing fine-grained rewards.

Building upon the above framework, we can effectively choose reasoning paths to conduct iterative Direct Preference Optimization (DPO) [7] and inference-time scaling. Experiments across multiple benchmarks demonstrate that our approach significantly enhances reasoning capabilities, outperforming strong baselines with consistent improvements. Our empirical analysis and ablations indicate that incorporating step-level evaluation is more reasonable and effective than relying solely on answer-level evaluation. Interestingly, we find that longer reasoning is not necessarily better for VLMs, and reasoning steps' quality is much more important than reasoning length. Different from prior arts, our work highlights the importance of introducing fine-grained reasoning chains in complex multimodal reasoning and leveraging fine-grained reward-based reinforcement learning to enhance VLMs. We hope our findings provide new insights for the community.

## 2 Related Work

**VLM Reasoning.** As the application of VLMs in complex tasks such as mathematics [30, 51] and science [61, 18, 31] grows, the ability to reason through complex problems becomes increasingly important. For instance, in solving mathematical diagram problems, Mavis [65] constructs a corresponding chain-of-thought (COT) [57] reasoning dataset and optimized the model through instruction fine-tuning [40]. Visual-CoT [42] focuses on enhancing the model's ability to correspond visual questions with key areas using CoT reasoning. Insight-V [10] designs a multi-agent system consisting of a reasoning agent dedicated to performing long-chain reasoning and a summary agent trained to judge and summarize reasoning results. The success of CoT in improving VLMs' complex visual reasoning reveals the importance of enabling the model to analyze intermediate reasoning steps in detail [66]. Unlike most existing methods [21, 42, 9] that use reasoning chains at a coarse-grained level, our work introduces a fine-grained structured reasoning approach, offering greater potential for enhancing the reasoning capabilities of VLMs.

**Inference-Time Scaling.** Inference time scaling [45] refers to the process wherein a model utilizes additional reasoning time to solve problems more effectively. This System-2 style thinking ability [39] is recognized as an effective way to enhance the complex reasoning capabilities of LLMs and

has recently garnered significant attention [69, 12]. Traditional inference time scaling methods include majority voting [14], best-N search [56], and sentence-level beam search [49]. Despite their widespread application, majority voting is typically only for problems with standard answers, whereas best-N search and sentence-level beam search present challenges in terms of accuracy assessment and the quality of individual responses [58]. Recently, OpenAI has developed verifiers to supervise and select reasoning paths during the inference process [25], while Math-Shepherd [52] evaluates intermediate reasoning steps based on their likelihood of leading to correct answers. Inspired by the success of these LLMs, LLaVA-CoT [58] and URSA [34] have started exploring the inference time scaling capabilities of VLMs. Procedural reasoning datasets have been constructed and used for the supervised fine-tuning of the models. Different from those prior arts, we utilize PRM to provide granular rewards and facilitate step-level beam search to enhance the inference-time scaling of VLMs.

# 3 Method

In this section, we present the pipeline of CoS as illustrated in Fig. 1. In Section 3.1, we introduce the design of our structured reasoning template. In Section 3.2, we describe the composition of the SFT dataset ShareGPT-Step-300K. Then we explain the acquisition of process annotation data in Section 3.3. Last, in Section 3.4, we elaborate on the details of the RL training pipeline with fine-grained rewards.

## 3.1 Structured Reasoning

We structure the entire thought into multiple consecutive cognitive steps, ensuring that these steps logically flow in a natural and coherent manner towards the final answer. Each step consists of the following three components:

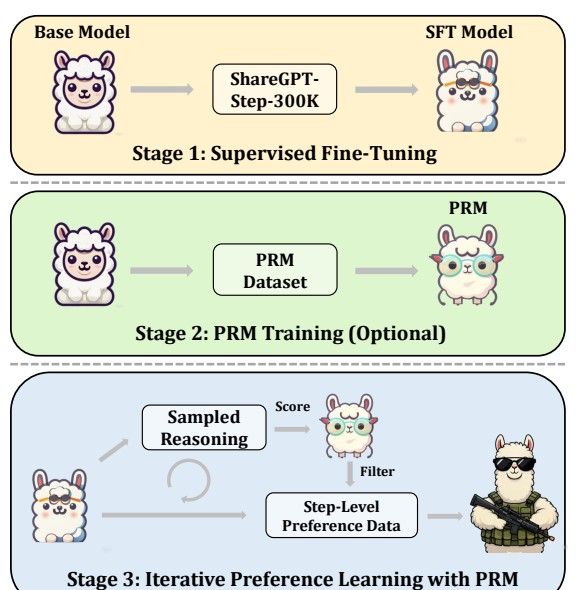

Figure 1: **Overview of our framework.** We conduct SFT to enable the model to generate step-by-step reasoning paths. Then we train a PRM to provide fine-grained RL reward, and enhance the complex reasoning ability through step-level preference learning. Note that PRM only needs to be trained once for different models.

- **Name.** A summary and overview of the current step, providing a high-level explanation. This outlines the key aspects of the step to be executed.

- **Thought.** A detailed description of the thought process in this step, explaining what is being observed, inferred, or analyzed.

- **Reflection.** Establishing connections between the visual content and previous steps. Since VLMs are prone to generating content that contradicts the image, the reflection is introduced to create links with the visual content and prior steps, thus alleviating hallucination.

We use special tokens to establish this reasoning format, rather than simply relying on additional prompts or external prompt engineering, because depending on prompt-based output format is unstable and would require more data filtering and cleaning. In contrast, ensuring stable output of this format is more critical for splitting and evaluating steps.

CoS begins with the special token $< |reasoning\_start| >$. We take reasoning step as the basic unit for structured and fine-grained reasoning. Steps are connected by $< |reasoning\_proceed| >$. The reasoning stops after outputting the token $< |reasoning\_end| >$, then the model derives the final answer from previous reasoning. The model generates reasoning steps and answers in an auto-regressive manner, where the number of reasoning steps and their specific content are self-generated without any restrictions. This structured output and its length are autonomously adapted and adjusted

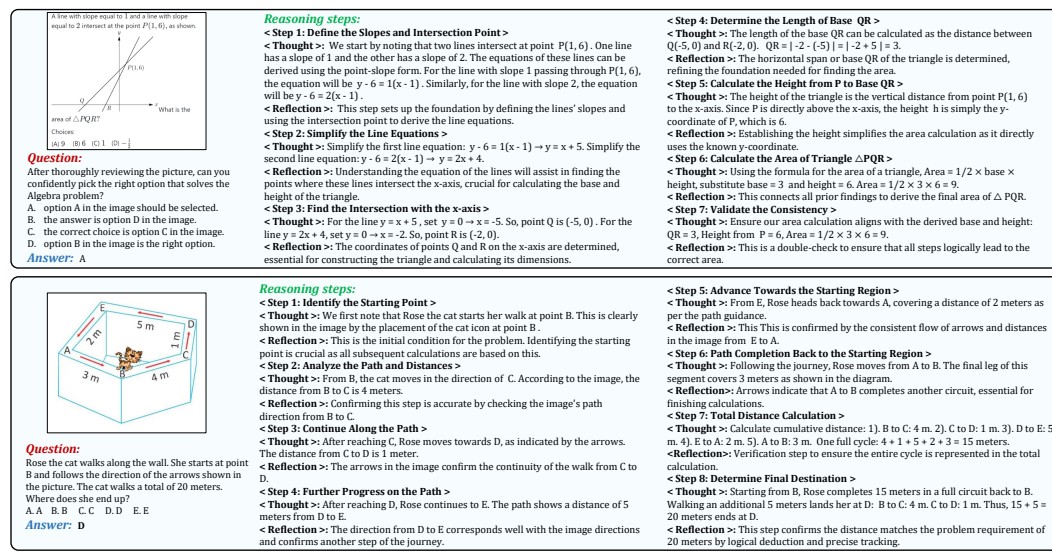

Figure 2: **Qualitative examples.** We depict the reasoning process on cases that need to understand both visual content and mathematics. Our CoS has accurate understandings of both aspects, and more importantly, combine them to derive multimodal reasoning. Note that special tokens are omitted.

by the model, thereby enhancing its ability to perform complex reasoning. We depict two qualitative examples in Fig. 2 and the details of special tokens can be found in Appendix A.

## 3.2 ShareGPT-Step-300K

Given that most VQA, multi-modal math, and reasoning datasets only provide short answer annotations, we design a method to leverage these short annotations to generate long, step-by-step reasoning data. Specifically, we provide the question and its ground-truth answers as references, and prompts GPT-4o to generate step-by-step reasoning. This can be seen as reasoning from the results. The reference answer can greatly reduce the difficulty of the question, making it easier to generate high-quality thinking steps.

To ensure diversity, our data is collected from various multimodal datasets, encompassing 17 tasks that require diverse multimodal reasoning capabilities. These tasks can be categorized into four major types including mathematical reasoning [51, 64, 3, 26, 24], scientific reasoning [18, 31, 23, 5], chart&document analysis [35, 32, 37, 36], and world knowledge [41, 54, 22, 44]. A summary of the dataset sources can be found in Appendix D.

After generating the data, we conducted meticulous data cleaning and filtering to remove a large amount of data that did not meet the required formatting standards. Ultimately, we obtained 300K high-quality step-level data instances, which we named as ShareGPT-Step-300K. We plan to release this dataset to advance research in the community on fine-grained step-level reasoning.

## 3.3 Step-Level Annotation

Currently, mainstream methods for evaluating reasoning steps can be categorized into three types [67]: **i)** Human, which relies on human annotators to produce high-quality data but incurs substantial costs; **ii)** Monte-Carlo (MC) Estimation, which infers the correctness of steps from the outcomes of a vast number of sampled solutions; **iii)** LLM-as-Judge, which leverages LLM as critic by directly prompting the LLM to evaluate the correctness of steps. In this work, for the sake of efficiency and scalability, we employ MC Estimation and LLM-as-Judge to annotate step-level data.

For MC Estimation, we use math-shepherd [52] to annotate steps automatically. Specifically, given a problem with a golden answer and a step-by-step solution, to annotate the correctness of a specific step, we utilize a fine-tuned VLM to decode N subsequent reasoning paths from this step. We further validate whether the decoded final answer matches with the golden answer. If a reasoning step can

deduce more correct answers than another, it would be assigned a higher correctness score. Here we use N=16 by default.

For LLM-as-Judge, we utilize GPT-4o [16] to perform the annotations. Each step is evaluated with one of three possible ratings: **i) Good**: the step is logical, directly relevant to the question, and most importantly, correct; **ii) Neutral**: the step is relevant but contains minor flaws or gaps; **iii) Bad**: the step is flawed, wrong, or shows a misunderstanding of the question or context. For solutions where the final answer is correct, both Good and Neutral ratings lead to a correct label. Conversely, for solutions with incorrect final answer, only steps annotated as Good are ultimately marked as correct. The prompt is provided in Appendix B.

We use both math-shepherd and GPT-4o to generate 100K step-annotated data, respectively. These step-annotated data are then utilized to train a PRM, which can provide fine-grained reward signals for reinforcement learning and inference-time scaling.

### 3.4 RL with Fine-grained Rewards

As illustrated in Fig. 1, our training pipeline is designed to be straightforward and efficient. Starting from a well-pretrained VLM, we first perform supervised fine-tuning on the ShareGPT-Step-300K dataset to enable the model to generate step-by-step reasoning paths. Subsequently, we train a PRM to provide fine-grained rewards. Leveraging the PRM, we efficiently select reasoning paths for iterative Direct Preference Optimization, thereby further enhancing the model's reasoning capabilities.

**Process reward model training.** In RLHF, the reward models are typically trained with binary preferences [2]. In reasoning tasks where step-annotation of solutions is accessible, reward models can be trained under the supervision of such ground-truth correctness. Given a question $q$, process reward model (PRM) estimates the correctness of individual reasoning steps. The PRM $r_{\text{process}}$ is trained with the following objective:

$$\mathcal{L}_{\textbf{PRM}} = \mathbb{E}_{q,p^{(k)},y_k \sim \mathcal{D}}[\text{Loss}(y_k, r_{\text{process}}(q, p^{(k)}))] \tag{1}$$

where $y^k$ is the label for the partial solution $p^{(k)}$ and Loss is the loss function. Here we adopt binary cross entropy loss by default. We train a InternVL-2.5-MPO-38B [55] on the step-annotated data as introduced in Section 3.3 to get our PRM. We ablate on the choice of PRM in Section 4.3.4.

**Iterative direct preference optimization.** Preference learning has gained increasing focus in VLMs [10]. To further enhance the quality of reasoning chains, we apply RL using the iterative DPO [7] algorithm to better align the model's reasoning process towards high-quality reasoning paths. We use SFT model as the policy model $\pi_\theta$, generating 16 reasoning paths for each question. Each reasoning path is evaluated with the trained PRM to assess its quality. We calculate the average score of reasoning steps and the score of final answer respectively, and take the weighted sum of them as the final score. The score is normalized to $[0, 1]$. Then we select positive-negative pairs depending on the scores. Note that the score of the positive path must exceed that of the negative path by a threshold $t$ to ensure sufficient distinction between them, thereby avoiding cases where the differentiation is minimal.

Formally, we denote the dataset as $\mathcal{D}_{\text{DPO}} = (x, y_+, y_-)$, where $x$ is the question, $y_+$ and $y_-$ are the positive and negative responses. The DPO objective is defined as below:

$$\mathcal{L}_{\textbf{DPO}}(\pi_\theta; \pi_{\text{ref}}) = -\mathbb{E}_{(x,y_+,y_-)\sim\mathcal{D}}\left[\log \sigma\left(\beta \log \frac{\pi_\theta(y_+|x)}{\pi_{\text{ref}}(y_+|x)} - \beta \log \frac{\pi_\theta(y_-|x)}{\pi_{\text{ref}}(y_-|x)}\right)\right] \tag{2}$$

where $\pi_\theta$ is the policy model to be optimized and $\pi_{\text{ref}}$ is the reference model and they are both initialized with SFT model. $\sigma$ is the logistic function and $\beta$ is set to 0.1. After training, we use the trained policy model $\pi_\theta$ to generate next-round data $\mathcal{D}_{\text{DPO}}^{(2)}$, while keeping $\pi_{\text{ref}}$ fixed (i.e., the SFT model). We repeat this procedure for multiple rounds to progressively enhance the the model's reasoning.

## 4 Experiment

In this section, we first introduce the setting and implementation details in Section 4.1. Then we present a comprehensive comparison with state-of-the-art VLMs across multiple reasoning benchmarks in Section 4.2. In Section 4.3, we conduct extensive ablation studies and analyses to validate the effectiveness of our approach and explore several interesting properties.

Table 1: **Comparison with open-source VLMs.** We evaluate our method on six benchmarks covering both general and task-specific reasoning capacities. Our CoS has consistently strong performances across these benchmarks, surpassing different baselines and other state-of-the-art VLMs by large margins. The items in **bold** and underlined respectively represent the first or second highest scores.

| Method | Size | MathVista | MMStar | MMMU | M3CoT | AI2D | ChartQA | Average |
|---|---|---|---|---|---|---|---|---|
| LLaVA-1.5 [27] | 13B | 27.6 | 32.8 | - | 39.5 | - | - | - |
| IXC-2.5 [63] | 7B | 63.7 | 59.9 | 42.9 | - | 81.5 | 82.2 | - |
| Ovis1.5-LLaMA3 [33] | 8B | 63.0 | 57.3 | 48.3 | - | - | 76.4 | - |
| LLaVA-CoT [58] | 11B | 54.8 | 57.6 | - | - | 78.7 | - | - |
| LlamaV-o1 [50] | 11B | 54.4 | 59.5 | - | - | 81.2 | - | - |
| Vision-R1-LlamaV [15] | 11B | 62.7 | 61.4 | - | - | - | 83.9 | - |
| R1-VL [62] | 7B | 63.5 | 60.0 | - | - | - | 83.9 | - |
| R1-Onevision [59] | 7B | 64.1 | - | - | - | - | - | - |
| MiniCPM-V-2-6 [60] | 8B | 60.6 | 57.5 | 49.8 | 56.0 | 82.1 | 79.4 | 64.2 |
| LLaVA-OneVision [21] | 7B | 63.2 | 61.7 | 48.8 | 52.3 | 81.4 | 80.0 | 64.6 |
| Qwen2-VL [53] | 7B | 58.2 | 60.7 | 53.7 | 57.8 | 83.0 | 77.4 | 65.1 |
| Insight-V [10] | 7B | 59.9 | 61.5 | 50.2 | 61.5 | 79.8 | 81.5 | 65.7 |
| InternVL-2.5 [6] | 8B | 62.2 | 59.6 | 54.1 | 62.4 | 84.5 | 84.9 | 68.0 |
| LLaVA-NeXT-LLaMA3 [28] | 8B | 45.9 | 43.1 | 36.9 | 45.6 | 71.5 | 69.4 | 52.1 |
| + SFT | 8B | 51.4 | 54.7 | 39.6 | 67.4 | 76.1 | 75.7 | 60.8 |
| **+ Iterative DPO (CoS-LLaVA)** | 8B | 54.7 | 58.9 | 41.8 | 71.7 | 79.2 | 79.1 | 64.2 |
| InternVL2.5-MPO [55] | 8B | 65.0 | 60.7 | 53.8 | 67.5 | 84.2 | 85.0 | 69.4 |
| + SFT | 8B | 65.9 | 61.0 | 53.7 | 75.7 | 81.6 | **88.3** | 71.0 |
| **+ Iterative DPO (CoS)** | 8B | **67.8** | **63.5** | **55.5** | **81.0** | **84.9** | 87.4 | **73.4** |

## 4.1 Experimental Setup

**Implementation details.** To verify the effectiveness and generalizability of our approach, we employ LLaVA-NeXt [28] and InternVL-2.5-MPO [55] as our base VLMs, respectively. For each base VLM, we first conduct one epoch of supervised fine-tuning on ShareGPT-Step-300K. Then, based on the SFT model, we apply three rounds of iterative DPO to obtain the final model. For each round, we compile approximately 20K preference pairs as training data for DPO. The training recipes can be found in Appendix C. For PRM, we select InternVL-2.5-MPO-38B [55] as the base model. We trained it for two epochs on 200K process-annotated data, using a learning rate of 1e-6 and a batch size of 128 to get the PRM. For the training time, SFT takes about 9 hours on a single A800 node, and three rounds of iterative DPO takes about 6 hours in total on a single A800 node.

**Benchmarks.** We evaluate our method on several representative benchmarks, spreading a wide range of tasks that demand the model's complex reasoning abilities. MathVista [30] is a well-known dataset for multi-modal reasoning, focusing on mathematics skills such as plane geometry, function and puzzle. MMMU [61] evaluates the model's multimodal reasoning capability across various university-level disciplines. ChartQA [35] aims at assessing logical reasoning with charts. MMstar [4] focuses on various multi-modal problems that must rely on visual contents. M3CoT [5] evaluates the model's chain-of-thought reasoning abilities. AI2D [18] is designed to test scientific knowledge and reasoning.

## 4.2 Main Results

We conducted a comprehensive comparison between our method and leading open-source VLMs on the aforementioned benchmarks. As shown in Table 1, our CoS achieves a substantial improvement over different baselines. Specifically, our method yields an average increase of 4.9% and 3.2% when applied to LLaVA-NeXT and InternVL-2.5-MPO, demonstrating the effectiveness and generalizability of our framework.

For LLaVA-NeXt, both SFT and DPO lead to significant performance gains, indicating that for relatively weaker VLMs, both SFT and reinforcement learning (RL) play crucial roles. In contrast, for InternVL2.5-MPO, the performance improvement from SFT alone is relatively limited. However, after applying iterative DPO, the model can achieve a notable performance boost, reaching state-of-the-art. This indicates that for stronger VLMs, the benefits of further improvement through SFT are marginal, while RL is more essential and effective for enhancing strong model's reasoning capabilities.

## 4.3 Analysis and Ablation Study

### 4.3.1 Reasoning Step Matters

Firstly, we dive into the impact of intermediate reasoning steps and final answers on evaluating reasoning quality. To be specific, we use LLaVA-NeXt [28] fine-tuned on the ShareGPT-Step-300K as base model and sample 16 reasoning paths and corresponding answers for each question in M3CoT. We then utilize a trained PRM to score these reasoning steps and answers. We use the weighted sum of the step score and answer score as the selection criterion to report best-of-16 accuracies.

As shown in Fig. 3, we progressively increase the weight of step scores from 0% (i.e., relying solely on the final answer for evaluation) to 100% (i.e., relying exclusively on reasoning steps). The model's best-of-16 accuracy exhibits a trend of initially increasing and then decreasing. This indicates that relying solely on either the final answer or the intermediate reasoning steps is not optimal; instead, a balanced evaluation that considers both is preferable. This conclusion also holds for reinforcement learning training as shown in Section 4.3.3. Unless mentioned, we set the weight of step scores to 20% by default in our experiments.

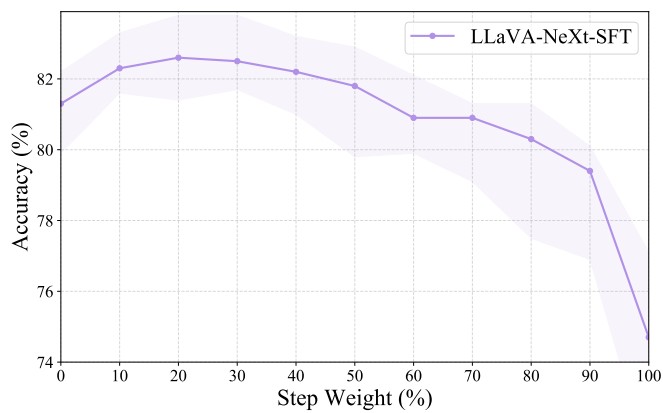

Figure 3: **Ablation on step weight.** We progressively increase the weight of step score from 0% to 100% and report the best-of-16 accuracy on M3CoT. Considering both step's and answer's quality leads to optimal performances.

### 4.3.2 Inference-Time Scaling

**Step-level beam search.** In order to further enhance the reasoning ability during inference, we design a step-level beam search by making full use of the structured output of CoS. The specific implementation steps are as follows:

- Sample N responses for the first step.
- Use PRM to score the N first steps.
- Continue to sample the next step based on the best step.
- Repeat the above steps until the final answer is generated.

It is worth noting that step-level beam search's cost and sample efficiency is exactly the same as that of Best-of-N (i.e., independently sample N responses, score with PRM, and select the best). An illustration is shown in Fig. 4.

Next, we explore the inference-time scaling properties of CoS reasoning. We compare four methods: Pass@N, Self-Consistency, PRM, and PRM with step-level beam search. Pass@N refers to the probability that at least one out of N sampled solutions is correct, which can be regarded as the upper bound of inference-time performance. Self-Consistency refers to selecting the most frequent answer among N sampled solutions. PRM refers to selecting the Best-of-N solution based on the scores provided by the PRM. PRM-BS refers to selecting the Best-of-N solution obtained through step-level beam search. We incrementally increase the number of samples N from 1 to 64 to evaluate the performance of these methods. As shown in Fig. 5, the answer selected by PRM has a significant advantage over Self Consistency. Moreover, PRM-BS has achieved consistent improvement over PRM. Considering that the sampling efficiency of PRM-BS and PRM is completely the same, this further illustrates the advantages of our method in the inference-time.

### 4.3.3 Ablation on RL training

We proceed to investigate the impact of rewards on the RL training. Specifically, we utilize LLaVA-NeXt-SFT as our base model and employ three methodologies to offline select training data for

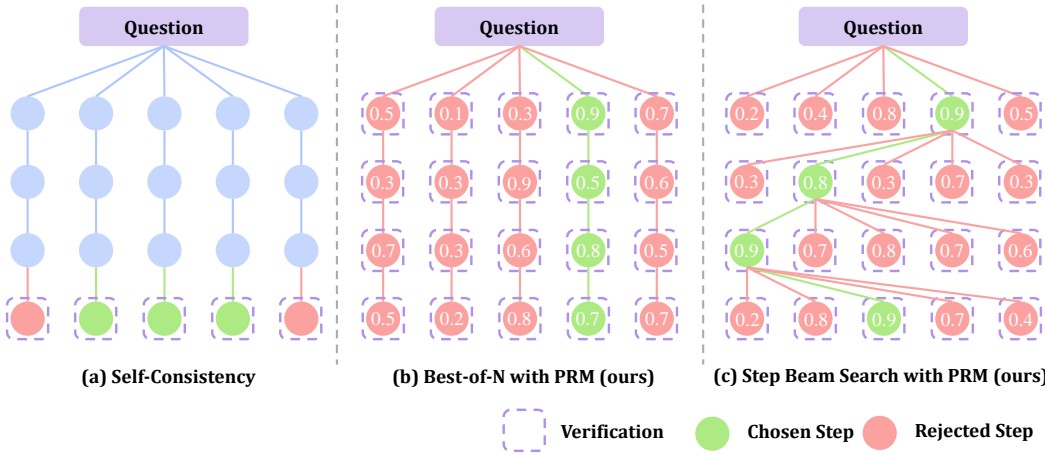

Figure 4: **Illustration of inference-time scaling.** Our structured and fine-grained designs enable us to execute (b) Best-of-N with PRM and (c) Step Beam Search with PRM. Both of them outperforms Self-Consistency by significant margins.

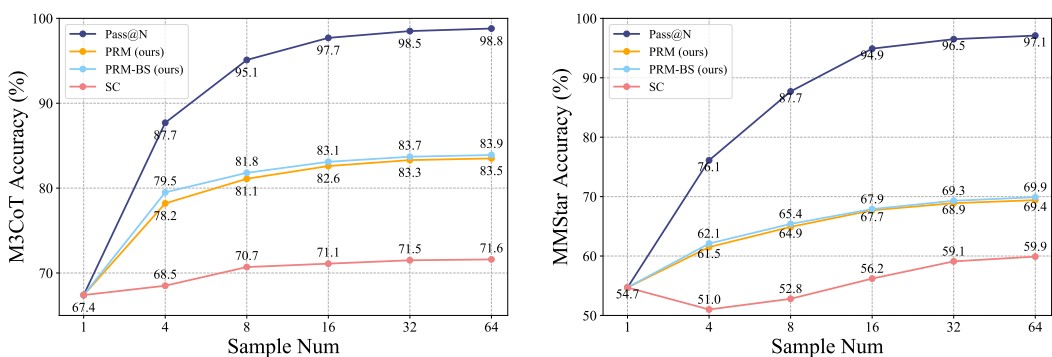

Figure 5: **Pass@N and Best-of-N comparison on M3CoT and MMStar.** Here SC refers to Self-Consistency, PRM-BS refers to step-level beam search. We increase N from 1 to 64, and our method demonstrates appealing inference-time property.

iterative DPO: **i)** Outcome. This approach uses the ground-truth labels of the questions as the gold criterion for judging the quality of responses. Responses with correct answers are selected as positive samples, while those with incorrect answers are negative ones; **ii)** PRM scores for answers. This can be seen as a lower bound on performance compared to the outcome method, as the accuracy of PRM's evaluation of answers is inherently lower than that of direct evaluation using ground-truth labels; **iii)** PRM scores for both reasoning steps and answers. This method comprehensively considers the quality of step and answer to selects positive and negative pairs.

As shown in Table 2, solely relying on answer scoring for selection underperforms the outcome-based method, which is an anticipated result. The strategy that takes into account both the intermediate steps and the answer quality achieves the best performances, surpassing outcome-based method with clear gains. This underscores the significance of fine-grained rewards in assessing reasoning quality, thereby enhancing the efficacy of RL training.

Table 2: **Ablation on the rewards for RL.** Taking both the steps and answer quality into account leads to the best performances.

| Method | MathVista | MMStar | M3CoT |
|---|---|---|---|
| LLaVA-NeXt-SFT | 51.4 | 54.7 | 67.4 |
| Answer (PRM) | 53.1 | 57.3 | 69.7 |
| Outcome | 53.5 | 58.1 | 70.0 |
| Step&Answer (PRM) | **54.7** | **58.9** | **71.7** |

Table 3: **Ablation on the PRM.** Stronger PRM leads to more accurate evaluation for reasoning steps.

| Base Model | Question | Step Acc | Answer Acc |
|---|---|---|---|
| LLaVA-NeXt-8B | Seen | 85.1 | 90.5 |
| LLaVA-NeXt-8B | Unseen | 83.7 | 84.7 |
| InternVL-2.5-8B | Unseen | 85.7 | 91.1 |
| InternVL-2.5-38B | Unseen | **87.3** | 88.7 |

### 4.3.4    Ablation on PRM

To evaluate the accuracy of the PRM, we reserved 10K process-annotated reasoning data for assessment. Among these, 5K questions were from the PRM training dataset, while the other 5K were unseen during PRM's training, allowing us to evaluate PRM's ability to assess in-domain data and generalization capability to unseen questions. We employed three base models: LLaVA-NeXt-8B, InternVL2.5-8B, and InternVL2.5-38B. As demonstrated in Table 3, the stronger the base model, the more accurate its judgment of intermediate steps for the PRM. Given the critical importance of step evaluation in our framework, we select InternVL2.5-38B as the base model for our PRM.

### 4.3.5    Reasoning Length

It has been observed that the length of reasoning tends to increase progressively when LLMs are trained with RL on mathematical problems, and their performance improves accordingly [12]. However, things become different for visual complex reasoning. Existing visual complex reasoning problems do not require lengthy intermediate processes as pure mathematics does, but rather rely on the utilization of visual information and the connection of relevant knowledge to trigger reasoning (e.g., geometric shapes). Therefore, we investigate how the length of VLM reasoning steps changes with training. We take LLaVA-NeXt-SFT as the base model and calculate the aver-

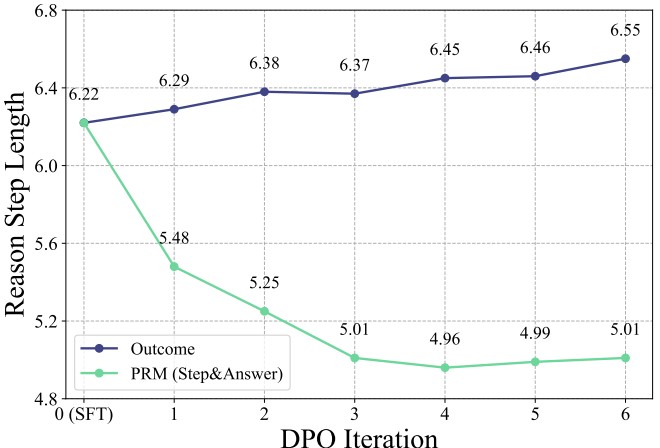

Figure 6: **Reasoning step length curve.** Increasing reasoning length is not necessarily better. For visual complex reasoning, ensuring high-quality steps is of vital importance.

age reasoning step after each round of outcome DPO and PRM DPO, respectively. We depict the change curve of reason step length in Fig. 6.

For outcome DPO, we select reasoning paths solely based on the correctness of the final results. It can be observed that as the training iteration progresses, the reasoning step length progressively increases. This indicates that the model tends to keep extending the reasoning length if we don't consider the quality of intermediate steps. Interestingly, the situation is entirely the opposite for PRM DPO. The model tends to initially reduce the length of reasoning to improve the quality of reasoning, and only after stabilizing does it slowly increase the length of reasoning. Considering that PRM DPO yields better results than outcome DPO as shown in Table 2, this phenomenon indicates that for visual complex reasoning, increasing reasoning length is not necessarily better. It is more crucial to first ensure that the model can output high-quality reasoning steps.

### 4.4    Ablation on Reasoning Pattern

In order to demonstrate our reasoning design's advantages, we add a direct quantitative comparison between the proposed method and simpler prompting strategies in Table below. Specifically, No Reason denotes a setting where the model performs direct question answering without any explicit reasoning; Direct Prompt means directly prompting VLMs to reason step by step (as in [19]); CoS corresponds to our proposed method using step-level structured reasoning with fine-grained rewards. All methods are trained using the same SFT and RL datasets to ensure a fair comparison. For No

Table 4: **Ablation on the reasoning pattern.** No Reason denotes a setting where the model performs direct question answering without any explicit reasoning; Direct Prompt means directly prompting VLMs to reason step by step. The corresponding SFT dataset's format has been adjusted accordingly to make a fair comparison.

| Method | Reason | Reward | MathVista | MMStar | M3CoT | Average |
|--------|--------|--------|-----------|--------|-------|---------|
| LLaVA-NeXt | - | - | 45.9 | 43.1 | 45.6 | 44.9 |
| No Reason SFT | ✗ | - | 49.1 | 53.9 | 62.1 | 55.0 |
| No Reason RL | ✗ | outcome | 51.5 | 56.4 | 63.4 | 57.1 (+2.1) |
| Direct Prompt SFT | ✓ | - | 51.2 | 54.9 | 66.3 | 57.5 |
| Direct Prompt RL | ✓ | outcome | 53.1 | 58.2 | 69.3 | 60.2 (+2.7) |
| CoS SFT | ✓ | - | 51.4 | 54.7 | 67.4 | 57.8 |
| CoS RL | ✓ | PRM | 54.7 | 58.9 | 71.7 | 61.8 (+4.0) |

Table 5: **Ablation for GRPO experiments.** Our Cos has consistently better performance than the outcome-reward counterpart.

| Method | Reward | MathVista | MMStar | M3CoT | Average |
|--------|--------|-----------|--------|-------|---------|
| LLaVA-NeXt-SFT | - | 51.4 | 54.7 | 67.4 | 57.8 |
| outcome GRPO | outcome | 54.3 | 57.9 | 71.4 | 61.2 |
| CoS GRPO | PRM | 56.3 | 59.1 | 73.7 | 63.0 |

Reason, we use only the question-answer pairs (without intermediate reasoning thought) from the ShareGPT-Step-300K dataset for SFT. For Direct Prompt, we concatenate all steps of reasoning (i.e., "thoughts") from each instance into a single reasoning sequence. Since both No Reason and Direct Prompt can not output strict step-level thought, we cannot assess their intermediate reasoning quality, so we apply the outcome reward (i.e., correctness of the final answer) during RL training.

There are two key observations: i) Models trained with reasoning-augmented SFT data (i.e., Direct Prompt and Cos) clearly outperform models trained with QA-only data (No Reason) on reasoning benchmarks; ii) Our Cos yields much greater improvements during RL compared to outcome-only reward settings. This ablation study directly supports our central motivation for fine-grained reasoning and strengthens the empirical foundation of our conclusions.

### 4.5 GRPO-based ablation

Recently, GRPO [43] has become the mainstream algorithm in current reinforcement learning and has demonstrated advantages across multiple tasks. Therefore, we further conducted an ablation experiments based on the GRPO algorithm to validate the effectiveness of our proposed method. For GRPO experiments, we deploy the 38B PRM as the reward server. We set batchsize to 128, rollout.n to 5, lr to 1e-5 and run 100 steps starting from the SFT model. As shown in Table 5, our fine-grained reward RL outperforms outcome reward RL consistently using GRPO, further strengthening our method and analysis.

## 5 Conclusion

In this work, we introduce Chain of Step (CoS) reasoning for vision-language models, enabling assessing reasoning step quality accurately and leading to effective reinforcement learning and inference-time scaling with fine-grained rewards. Experimental results across multiple benchmarks demonstrate the effectiveness of CoS. More importantly, we conduct extensive empirical analysis and ablations to unveil CoS's appealing properties. We hope this paper offers insights into more complex multi-modal reasoning.

## Acknowledgments

This work is supported in part by the National Science and Technology Major Project, Grant No.2022ZD0116403.

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

Figure 7: **Illustration of structured reasoning.** We take reasoning step (special tokens marked in blue) as the basic unit to conduct structured and fine-grained reasoning.

## A  Special Tokens

Our CoS uses special tokens to realize structured step-by-step reasoning. We list all 11 special tokens and their corresponding definitions in Table 6. For the initialization of these special tokens, we use the average value of the token embeddings of its corresponding definition. The reasoning structure is illustrated in Fig. 7.

## B  Prompts

In this section, we provide a demonstration of the prompts used for SFT and PRM data construction. The prompt used to create ShareGPT-Step-300K is illustrated in Fig. 8. The prompt used to utilize GPT-4o to label process-level annotation is depicted in Fig. 9.

## C  Training Recipes

In this section, we provide the specific hyper-parameter settings for the three stages in Table 7. All SFT and iterative DPO experiments are conducted on $8 \times$ NVIDIA-A800 GPUs by default. For PRM, it is trained on $16 \times$ NVIDIA-A800 GPUs. Additionally, we provide important parameters used in data construction. During the generation of positive and negative example pairs and the Math-Shepherd [52] Sampling period, we set the temperature to 1.0, n return sequences to 16, and top p to 0.95.

Table 6: **Special tokens we utilized and their definitions.**

| Special token | Definition |
|---|---|
| `<|reasoning_start|>` | Start token of the reasoning process |
| `<|reasoning_end|>` | End token of the reasoning process |
| `<|reasoning_step_start|>` | Start token of a single reasoning step |
| `<|reasoning_step_end|>` | End token of a single reasoning step |
| `<|reasoning_step_name_start|>` | Start token of the reasoning step name |
| `<|reasoning_step_name_end|>` | End token of the reasoning step name |
| `<|reasoning_step_thought_start|>` | Start token of the reasoning thought content |
| `<|reasoning_step_thought_end|>` | End token of the reasoning thought content |
| `<|reasoning_step_reflection_start|>` | Start token of the reasoning reflection content |
| `<|reasoning_step_reflection_end|>` | End token of the reasoning reflection content |
| `<|reasoning_proceed|>` | Delimiter token between reasoning steps |

Table 7: **Hyperparameter setting and training recipes.**

| Hyper-parameter | SFT | PRM Training | DPO |
|---|---|---|---|
| Learning Rate | 5e-6 | 1e-6 | 1e-6 |
| Epoch | 1 | 2 | 3 (iteration) |
| Warm-up Ratio | 0.03 | 0.03 | 0.03 |
| Weight Decay | 0.05 | 0.05 | 0.05 |
| Batch Size | 128 | 128 | 256 |
| Drop-path Rate | 0.1 | 0.4 | 0.1 |
| Update Parts | Projector, LLM | Projector, LLM | ViT, Projector, LLM |
| Data Size | 300K | 200K | 20K |

## D    Dataset Sources

To ensure diversity, our data is collected from various multimodal datasets, encompassing 17 tasks that require diverse multimodal reasoning capabilities. As shown in Table 8, these tasks can be categorized into four major types including mathematical reasoning [51, 64, 3, 26, 24], scientific reasoning [18, 31, 23, 5], chart&document analysis [35, 32, 37, 36], and world knowledge [41, 54, 22, 44].

Table 8: **SFT dataset sources.** To ensure diversity, we collect our dataset from a wide range of tasks that demands diverse reasoning capabilities.

| Task Type | Dataset |
|---|---|
| Mathematics | GeoQA+ [3], MathVision [51], SceMQA [24], MathVerse [64], CLEVER-Math [26] |
| Chart | ChartQA [35], TabMWP [32] |
| Science | ArxivQA [23], M3CoT [5], ScienceQA [31], AI2D [18] |
| Document | DocVQA [37], InfoVQA [36] |
| Knowledge | A-OKVQA [41], FVQA [54], TextVQA [44], SEED [22] |

## E    Attempt for Step-wise DPO

Due to our structured and fine-grained design, we can improve the reasoning ability of the model by conducting fine-grained step level beam search during reasoning. Naturally, we wonder whether we could construct similar fine-grained preference data during RL training. Specifically, we use PRM to find the optimal step and a bad one at each step, and then build a set of pair data. Then we sample the next step based on the optimal step, and cycle until we get the final answer. In this way, we build a pair of preference data at each step, which can provide step-wise fine-grained preference rewards. We term this method to be step-wise DPO. This is different from step-dpo [20], which only detect the first error step while we build pairs of every single step.

However, this method does not work as expected, but has a significant negative impact on model performances as shown in Table 9. We believe that the failure of this method lies in that the next sampling based on a good step is probably also a good step. Therefore, the distinction between the constructed chosen and reject pairs is relatively small. As a consequence, in order to avoid outputting negative steps during training, the model will also tend not to output chosen steps. This conjecture is validated in the rewards: the rewards of chosen and reject are both very small negative numbers. Therefore, we mainly conduct fine-grained step-by-step sampling in the inference time.

## F    Discussion of Table 3

We observed a counterintuitive phenomenon that the larger InternVL-2.5-38B does not achieve a higher Answer Accuracy compared to its smaller counterparts. Our speculation on this phenomenon is as follows:

Table 9: **Ablation on per-step-wise DPO.** The chosen and rejected steps are too similar to form effective preference pair, so the model will refuse to output both, backfiring performances.

| Method | M3CoT | MathVista | Chosen Rewards | Reject Rewards |
|---|---|---|---|---|
| SFT | 67.4 | 51.4 | - | - |
| Outcome | 70.0 | 53.5 | -0.07 | -0.16 |
| Per-step-wise | 64.8 | 50.7 | -1.39 | -3.86 |

Notably, evaluating intermediate reasoning steps is substantially more challenging than evaluating the final answer for three reasons: i) Higher semantic complexity: Step-level evaluation requires a deeper and more holistic understanding of the question. The model must assess whether each step contributes constructively toward the final answer. Similarly, annotating process data is also more cognitively demanding for human annotators. ii) More supervision signals. As shown in Equation 1, PRM is trained to minimize the sum of cross-entropy losses over all intermediate steps and the final answer. A complete reasoning trace contains multiple steps but only a single answer, thus accurately assessing all the intermediate steps is much harder than only assessing the answer. iii) Answer memorization. As highlighted in recent studies [13, 46], LLMs often exhibit "answer hacking"—producing correct answers via memorized question or patterns rather than genuine understanding. In contrast, intermediate steps exhibit higher variance across instances and are less amenable to memorization.

As we know, neural networks are inherently short-cut learners[11]. During training, smaller models (i.e., 8B PRM) tend to prioritize optimizing for final answer accuracy, as it offers a more accessible gradient path to loss reduction. Improving step-level predictions poses a greater optimization challenge. The 38B PRM, with its higher capacity, is better able to model and optimize intermediate reasoning quality. Consequently, it achieves greater improvements in step-level accuracy, which more effectively reduces the total loss—resulting in the pattern observed in Table 3.

# G   Limitations

We generated process-labeled data through Monte Carlo estimation and LLM-as-Judge and verified the effectiveness of our method through extensive experiments. But it cannot be denied that the process data obtained by these two methods still cannot be guaranteed to be 100% correct. Even the process data labeled by humans cannot guarantee accuracy and quality [25], and we do not use human annotation due to the high cost and inability to scale. In the future, if there is a way to produce higher-quality process labeled data, our method can hopefully shine further.

# H   Broader Impacts

This paper presents work whose goal is to advance the field of Vision-language Models and Machine Learning. There are many potential societal consequences of our work, none of which we feel must be specifically highlighted here.

You are a thought assistant, i will provide you with an image, a question, and a reference answer. Your responsibility is to develop a sequence of intermediate thinking steps that would logically connect the question to its answer.

Each step must consist of and only consist of the following three parts:
1. **Step Name**: A brief title summarizing the main idea of the step.
2. **Detailed Thinking:** An in−depth description of the thought process in this step, explaining what is being observed, inferred, or analyzed. Note that the process of thinking and reasoning should be presented in as much detail as possible.
3. **Reflection:** A reflection on this step's connection to prior steps, pointing out any potential errors, and if possible, citing evidence in the image that supports or informs this step.

There are some guidelines you should follow:
1. Make sure the steps are clear, sequential, and detailed enough to make each step flows logically into the next.
2. The number of steps must be no less than 5 and no more than 10.
3. Do not state an answer at the beginning. Explain the visual clues that help to derive the answer.
4. Don't state that the reference answer is correct or consistent to your finding. You are writing your own solution.
5. The explanation can help you derive the thinking process.
6. For math problems, you must give specific operation steps and concrete intermediate solution leading to the final answer.
7. Make sure to incorporate concrete observations from the provided materials.

The output format must be strictly organized as follows:
### <Step 1: xxx>
<Detailed Thinking>: xxx
<Reflection>: xxx

### <Step 2: xxx>
<Detailed Thinking>: xxx
<Reflection>: xxx

...

### <Step x: xxx>
<Detailed Thinking>: xxx
<Reflection>: xxx

...

### <Answer>
xxx

Figure 8: Prompt used for generating step-by-step reasoning data.

You are a judging assistant specializing in evaluating process reward, the following is an image, a question, a reference answer and a step-by-step intermediate answer process for the question. The reference answer is correct while the intermediate answer steps are not necessarily correct.

Your task is to review and critique each step of the intermediate answer process based on its correctness. Assign one of the following ratings to each step:

- **Good:** The step is logical, directly relevant to the question, and most importantly, correct. This step provides substantive progress to solve the question.
- **Neutral:** The step is somewhat relevant but contains minor flaws or gaps in reasoning. Or this step only includes some abstract practices rather than concrete solutions, which is not helpful for solving the question in effect.
- **Bad:** The step is flawed, wrong, or shows a misunderstanding of the question or context. This step has a negative effect on solving the question.

For each step, provide a brief explanation of your evaluation, identifying areas for improvement if Neutral or Bad. Finally, return your output in the form of a step label list. 1 stands for Good; 0 stands for Neutral; -1 stands for Bad.

The final step label list format must be strictly organized as follows:
### Step Label: [1,1,0,-1,-1,-1]

Figure 9: Prompt used for generating process-level annotation.

