# OpenReview forum: "Unveiling Chain of Step Reasoning for Vision-Language Models with Fine-grained Rewards"
_NeurIPS.cc/2025/Conference — NeurIPS 2025 poster_

### Official Review · Reviewer_iMMk · 2025-06-20

**Clarity:** 3
**Significance:** 3
**Originality:** 2
**Rating:** 4
**Confidence:** 4

**Summary:**

This work presents Chain-of-Step (CoS) reasoning for vision-language models (VLMs), introducing a structured, step-wise reasoning framework with fine-grained step-level rewards via a Process Reward Model (PRM). The authors construct the ShareGPT-Step-300K dataset using GPT-4o, develop the PRM for step-level evaluation, and integrate both into a direct preference optimization (DPO) training pipeline. Evaluation on several VQA benchmarks shows that this approach outperforms existing state-of-the-art VLMs.

**Questions:**

Please answer my concerns in weakness.
Besides, I have two additional questions that I believe need to be discussed.
1. Some works [2,3] indicate that using LLM-based reward models can introduce biases, which is termed as “reward hacking”. Does this also happen as you use VLM as PRM?
2. In Table 3, why does InternVL-2.5-38B, the PRM, achieve the highest Step Acc but only the third-highest Answer Acc?

[2] Liu, et al. LLMs as Narcissistic Evaluators: When Ego Inflates Evaluation Scores. ACL (findings) 2024.

[3] Wang, et al. Large Language Models are not Fair Evaluators. ACL 2024.

**Ethical Concerns:**

["NO or VERY MINOR ethics concerns only"]

**Final Justification:**

After carefully reading the rebuttal and discussing it with the authors, I raised my score from 2 to 4.
* Why not higher score?
    1. The presentation of this paper does not highlight its contribution.
    2. The writing of this paper, especially the experiment section, requires to be improved.

* Why not lower score?
    1. The proposed iterative DPO is technically novel, since it provides another way to empower MLLM with reasoning capability without using GRPO-based RL.
    2. During rebuttal, the authors solved my concerns about the experiment setting and provided the experiment results.

**Limitations:**

yes

**Quality:**

2

**Strengths And Weaknesses:**

**[Strengths]**

1. The idea of moving from traditional, coarse-grained chain-of-thought (CoT) reasoning to a fine-grained, explicitly structured process is well-motivated.
2. The selected benchmarks are extensive.

**[Weaknesses]**

1. It remains unclear whether directly prompting VLMs to reason step by step (as in [1]) could achieve a comparable level of structured thinking. This paper lacks a direct quantitative comparison between the proposed method and simpler prompting strategies. Including such a baseline would clarify the added value of the structured thinking.
2. The use of GPT-4o to generate the ShareGPT-Step-300K dataset is weird. In my understanding, GPT-4o is not known for fine-grained reasoning; it is unclear how it can generate high-quality, step-wise data as required by your framework.
3. The PRM is built using a much larger model (InternVL-2.5-MPO-38B) than the reasoning model (RM, e.g., 8B). However, in practical applications, the PRM should be constrained to be the same size as, or smaller than, the RM (Please correct if I am wrong). Thus, I expect that the Table 1 should be re-evaluated, since the 38B model is used as PRM.


[1] Kojima, Takeshi, et al. Large language models are zero-shot reasoners. NeurIPS 2022.

---

> ### Author Rebuttal · Authors · 2025-07-28
>
> Dear reviewer iMMK,
>
> Thank you for your feedback and help in improving our work. Below we have responsed to every question individually. In the discussion phase, please enlighten us on anything we can do to further improve the paper.
>
> **Weakness 1**
>
> Thank you for the insightful suggestion. We add an additional ablation study in the table below. Specifically, No Reason denotes a setting where the model performs direct question answering without any explicit reasoning; Direct Prompt means directly prompting VLMs to reason step by step (as in [1]); Cos corresponds to our proposed method using step-level structured reasoning with fine-grained rewards. All methods are trained using the same SFT and RL datasets to ensure a fair comparison. For No Reason, we use only the question-answer pairs (without intermediate reasoning thought) from the ShareGPT-Step-300K dataset for SFT. For Direct Prompt, we concatenate all steps of reasoning (i.e., "thoughts") from each instance into a single reasoning sequence. Since both No Reason and Direct Prompt can not output strict step-level thought, we cannot assess their intermediate reasoning quality, so we apply the outcome reward (i.e., correctness of the final answer) during RL training.
>
> There are two key observations: **i)** Models trained with reasoning-augmented SFT data (i.e., Direct Prompt and Cos) clearly outperform models trained with QA-only data (No Reason) on reasoning benchmarks; **ii)** Our Cos yields much greater improvements during RL compared to outcome-only reward settings. This ablation study directly supports our central motivation for fine-grained reasoning and strengthens the empirical foundation of our conclusions.
>
> |                   | Reason | RL Reward | MathVista | MMStar | M3CoT | Average     |
> | ----------------- | ------ | --------- | --------- | ------ | ----- | ----------- |
> | LLaVA-NeXt        | -      | -         | 45.9      | 43.1   | 45.6  | 44.9        |
> | No Reason SFT     | ×      | -         | 49.1      | 53.9   | 62.1  | 55.0        |
> | No Reason RL      | ×      | outcome   | 51.5      | 56.4   | 63.4  | 57.1 (+2.1) |
> | Direct Prompt SFT | √      | -         | 51.2      | 54.9   | 66.3  | 57.5        |
> | Direct Prompt RL  | √      | outcome   | 53.1      | 58.2   | 69.3  | 60.2 (+2.7) |
> | CoS SFT           | √      | -         | 51.4      | 54.7   | 67.4  | 57.8        |
> | CoS RL            | √      | PRM       | 54.7      | 58.9   | 71.7  | 61.8 (+4.0) |
>
> **Weakness 2**
>
> We chose to use GPT-4o to construct the Share-GPT-300K dataset for three main reasons: **i)** The primary objective of our SFT stage is not to improve the model's reasoning ability. The core goal of this paper is to explore step-level reasoning driven by fine-grained reward signals. To enable this, we first require the model to generate reasoning in a structured, step-wise format. Therefore, the main purpose of constructing the SFT dataset is to guide the model towards producing step-level reasoning trajectories that facilitate fine-grained reward assessment. At this stage, the key requirement is to generate coherent and logically sound intermediate reasoning steps; **ii)** Ground-truth is provided as reference. In order to ensure high-quality intermediate reasoning, we provide ground-truth answers as references when prompting GPT. This fundamentally alters the difficulty of the task, making the difference of reasoning abilities between 4o and stronger reasoning models (e.g., o1) less important . In such a setup, instruction-following ability and output consistency become more critical. In our preliminary experiments, when GT answers were provided as hints, GPT-4o did not underperform GPT-o1 in terms of reasoning quality obviously. On the other hand, GPT-4o produced more stable outputs and demonstrated stronger adherence to the target format; **iii)** Money cost. Using GPT-o1 would be more than 10× more expensive than GPT-4o, making the cost of money unaffordable. We also use GPT-4o to generate process annotation for the latter two reasons—better instruction-following and significantly lower cost.
>
> **Weakness 3**
>
> Thanks for the suggestion. The reason why we use a 38B PRM is that we hope to provide as accurate process rewards as possible, so that our exploration and empirical studies on fine-grained reward driven reasoning can be more precise and solid. Therefore, we use the largest model supported by our computing resources as the PRM. To address your concern, we conduct a further ablation on the rewards in RL using the 8B-PRM (i.e., expanding Table 2 in the paper). Compared with 38B-PRM, 8B-PRM has a lower accuracy of steps and a higher accuracy of answers as shown in Table 3 in the paper. Despite its lower step accuracy, the results support the same conclusion: jointly optimizing rewards for both intermediate reasoning and final answers yields superior performance. Therefore, changing the PRM of 38B to 8B will not significantly undermine the performance in Table 1, nor will it affect our core conclusions and findings.
>
> | Reward                                             | MathVista | MMStar | M3CoT | Average |
> | -------------------------------------------------- | --------- | ------ | ----- | ------- |
> | LLaVA-NeXt-SFT                                     | 51.4      | 54.7   | 67.4  | 57.8    |
> | Answer (PRM-38B)                                   | 53.1      | 57.3   | 69.7  | 60.0    |
> | Answer (PRM-8B)                                    | 53.3      | 57.3   | 70.1  | 60.2    |
> | Outcome                                            | 53.5      | 58.1   | 70.0  | 60.5    |
> | Step&Answer (PRM-38B) (i.e., Cos-LLaVA in Table 1) | 54.7      | 58.9   | 71.7  | 61.8    |
> | Step&Answer (PRM-8B)                               | 54.6      | 58.3   | 71.5  | 61.5    |
>
> **Question 1**
>
> Currently, we have not observed obvious reward hacking phenomenon in our experiments.
>
> **Question 2**
>
> While the results in Table 3 may appear counterintuitive, they are in fact consistent with the underlying training dynamics.
>
> Notably, **evaluating intermediate reasoning steps is substantially more challenging than evaluating the final answer** for three reasons: **i)** Higher semantic complexity: Step-level evaluation requires a deeper and more holistic understanding of the question. The model must assess whether each step contributes constructively toward the final answer. Similarly, annotating process data is also more cognitively demanding for human annotators[2]. **ii)** More supervision signals. As shown in equation 1 in the paper, PRM is trained to minimize the sum of cross-entropy losses over all intermediate steps and the final answer. A complete reasoning trace contains multiple steps but only a single answer, thus accurately assessing all the intermediate steps is much harder than only assessing the answer. **iii)** Answer memorization. As highlighted in recent studies[3,4], LLMs often exhibit "answer hacking"—producing correct answers via memorized question or patterns rather than genuine understanding. In contrast, intermediate steps exhibit higher variance across instances and are less amenable to memorization.
>
> As we know, **neural networks are inherently short-cut learners**[5]. During training, smaller models (i.e., 8B PRM) tend to prioritize optimizing for final answer accuracy, as it offers a more accessible gradient path to loss reduction. Improving step-level predictions poses a greater optimization challenge. The 38B PRM, with its higher capacity, is better able to model and optimize intermediate reasoning quality. Consequently, it achieves greater improvements in step-level accuracy, which more effectively reduces the total loss—resulting in the performance pattern observed in Table 3.
>
>
>
> [1] Large language models are zero-shot reasoners.
> [2] Let's Verify Step by Step.
> [3] SoK: Memorization in General-Purpose Large Language Models.
> [4] Rethinking Memorization in LLMs: On Learning by Rote vs. with Understanding.
> [5] Shortcut Learning in Deep Neural Networks.

---

> > ### Comment · Reviewer_iMMk · 2025-08-05
> > **Official Response by Reviewer iMMk**
> >
> > Thanks for the comprehensive rebuttal with additional experiments. Most concerns are resolved, so I'm raising my rating to borderline reject.  The remaining concern is the step-acc vs answer-acc discrepancy, which contradicts the paper's core idea, which uses fine-grained step reasoning to improve performance made by previous coarse-grained reasoning approaches.

---

> ### Comment · Area_Chair_uewq · 2025-08-05
>
> Hi,
> As the deadline of author-reviewer discussion is approaching, could you please check the authors' rebuttal and respond accordingly?
>
> Thanks
> AC

---

> ### Author Response · Authors · 2025-08-05
> **Further Response**
>
> Dear Reviewer iMMK,
>
> Thanks for replying and raising your rating to 3. For your last remaining concern, there exists a key misunderstanding that we would like to clarify.
>
> The step-acc v.s. answer-acc discrepancy (i.e., Table 3 in our paper) is a phenomenon specific to the PRM, while our core idea is  studying the fine-grained reasoning of the **actor** model. Research on **actor model** and **reward model** belongs to two different subfields, and our work falls into the former. The PRM, as a reward model, can be viewed as a binary classification model that outputs the probability of a step or answer being labeled as "good" or "bad." Given that the PRM does not perform reasoning at all but merely outputs classification scores, we respectfully differ that its classification discrepancy could not contradict the actor model's reasoning pattern conclusion.
>
> Actually, we found this phenomenon of PRM counter-intuitive but interesting, which is why we reported it in the paper for open discussion. In our response to Question 2, we provided our interpretation of this phenomenon. We understand that you may reckon the explanation not convincing enough, but what we want to emphasize is that **this specific phenomenon of reward model does not contradict our core idea and conclusion about the actor model**. As shown in the experiments (our response to weakness 3), using PRMs with accuracy discrepancy (i.e., 8B vs. 38B) does not alter the core findings and conclusions of our study. The important thing is to use PRM to provide rewards to both intermediate steps and the final answer, while minor differences in the step or answer accuracy are less important.
>
> In summary, this paper is a study of actor model's fine-grained reasoning, and PRM is just a tool to provide reward signals. If you think that the report of this counter-intuitive phenomenon is likely to lead readers to similar misunderstandings, it's okay for us to delete Table 3 without affecting our core conclusions and findings.
>
> Great thanks for your engagement and suggestions to improve our paper, and we sincerely hope the above clarification can eliminate your last remaining concern and raise to a positive rating. If there is anything further we can do or clarify, please let us know.

---

> > ### Comment · Reviewer_iMMk · 2025-08-07
> > **Follow-up Response from Reviewer iMMk**
> >
> > After a thorough re-reading of the paper, I realized that I had initially misunderstood some aspects of the work. I now appreciate that a central contribution of this paper is the use of iterative DPO to empower VLMs with reasoning abilities, without relying on GRPO-based reinforcement learning. This presents a valuable and novel perspective for the community. Given this insight, I agree that the minor differences in steps or answer accuracy are less important. Accordingly, I am raising my final score to 4.
> >
> > That said, I encourage the authors to revise the experimental sections (instead of removing Table 3) to include more detailed discussion and analysis. This would help mitigate potential misunderstandings for future readers and further clarify the contributions of the work.

---

> > > ### Author Response · Authors · 2025-08-07
> > > **Further Response**
> > >
> > > Dear Reviewer iMMK,
> > >
> > >   Great thanks for your time and suggestions! We will revise the paper as suggested in the final version.

---

### Official Review · Reviewer_zrV1 · 2025-06-29

**Clarity:** 4
**Significance:** 3
**Originality:** 3
**Rating:** 5
**Confidence:** 5

**Summary:**

The paper introduces Chain of Step (CoS) reasoning for vision-language models (VLMs), designed to enable accurate assessment of reasoning step quality and better reasoning capabilities. The core innovation is to propose a step-level reasoning dataset, PRM, and iterative DPO training method. Based on the proposed dataset and method, the author validate the effectiveness on different base VLMs and reasoning benchmarks. In addition, exhaustive ablation studies are conducted to analyze each components, and draw some interesting conclusions (e.g., PRM leads to shorter reasoning length but better performance).

**Questions:**

1. How will the method perform if GPT-4o is replaced with an open-sourced model?

**Ethical Concerns:**

["NO or VERY MINOR ethics concerns only"]

**Final Justification:**

The rebuttal addressed my question. Therefore, I maintain my rating as accept.

**Limitations:**

yes

**Quality:**

4

**Strengths And Weaknesses:**

# Strengths
1. The CoS method is simple and effective. With the proposed method, LLaVA-NeXT-LLaMA3 can get a over 10 points on the average of 6 reasoning benchmarks. I believe the dataset and method will provide a good baseline for the VL community.
2. The paper's comprehensive ablation studies yield crucial insights into VLM reasoning. Specifically, they highlight the critical importance of individual reasoning steps and emphasize that the quality of steps outweighs their length. These findings are invaluable for the future development of VL reasoning models.
3. The clarity and conciseness of the method's illustration are commendable. I really like the writing style: presenting each component with remarkable simplicity and directness, which greatly aids in comprehension.

# Weakness
1. The method seems to be dependent on GPT-4o. It remains unclear how the proposed approach would perform if a less powerful VLM was substituted for GPT-4o in its key components.

---

> ### Author Rebuttal · Authors · 2025-07-28
>
> Dear reviewer zrV1,
>
> Thank you for your feedback and help in improving our work. Below we have responsed to every question individually. In the discussion phase, please enlighten us on anything we can do to further improve the paper.
>
> **Weakness 1 & Question 1**
>
> We utilize GPT-4o in two components of our pipeline:
>
> 1. **SFT dataset construction**. A key goal of this work is to explore step-level reasoning driven by fine-grained rewards. To enable this, we first require the model to generate outputs in a structured, step-level format. The SFT dataset is constructed to guide the model toward this output format. At this stage, the primary requirement is that the teacher model produces reasonably coherent targeted step-level reasoning traces; high performance is not critical. Thus, GPT-4o can be replaced by a moderately capable open-source VLM (e.g., Qwen-VL) without significantly affecting overall performance, as the purpose of SFT is to induce the output structure rather than to boost reasoning quality.
> 2. **Step-level process annotation**. This component relies on GPT-4o to evaluate the quality of intermediate reasoning steps. This requires a model with strong reasoning capability, precise instruction-following, and the ability to produce consistent judgments in a target output format. In this case, replacing GPT-4o with weaker open-source models would substantially degrade annotation accuracy, which in turn would negatively impact the training of the PRM and compromise the reliability of the empirical findings.

---

> > ### Comment · Reviewer_zrV1 · 2025-08-08
> >
> > Thanks for providing the explanation. It addressed my question. I will keep my rating unchanged.

---

### Official Review · Reviewer_gk2j · 2025-07-03

**Clarity:** 3
**Significance:** 3
**Originality:** 2
**Rating:** 4
**Confidence:** 3

**Summary:**

This paper introduces Chain-of-Step (CoS) reasoning, a framework designed to enhance the fine-grained reasoning capabilities of vision-language models (VLMs). The approach decomposes reasoning into structured steps with three components (Name, Thought, Reflection) and leverages a 300K dataset (ShareGPT-Step-300K) for supervised fine-tuning (SFT). It further proposes a Process Reward Model (PRM) trained on step-annotated data to provide fine-grained rewards, enabling effective reinforcement learning via iterative Direct Preference Optimization (DPO). Experiments across multiple benchmarks (e.g., MathVista, MMStar) demonstrate consistent improvements over baselines.

**Questions:**

1. It appears that the authors have re-implemented supervised fine-tuning (SFT) and Direct Preference Optimization (DPO)-based reinforcement learning from human feedback (RLHF) on vision-language models (VLMs). From a novelty perspective, what are the essential differences between this work and the application of SFT and DPO in large language models (LLMs)?
2. The baselines only include models size smaller than 8B such as Qwen2-VL-7B, without comparing against the results of the latest Qwen2.5-VL (7B, 72B).
3. The base model used is only an 8B model. It would be interesting to explore how the proposed approach would perform with larger models, such as Qwen2.5-VL 72B.
4. After defining the step name, what is the basis for splitting a task into specific steps when completing it?
5. It would be valuable to see the performance of the proposed approach across additional benchmarks, including MathVerse[1], MathVisoin[2], MMMU-Pro[3], EMMA[4], MegaBench[5].


- [1] Renrui Zhang, Dongzhi Jiang, Yichi Zhang, Haokun Lin, Ziyu Guo, Pengshuo Qiu, Aojun Zhou, Pan Lu, Kai-Wei Chang, Yu Qiao, et al. Mathverse: Does your multi-modal llm truly see the diagrams in visual math problems? In European Conference on Computer Vision, pages 169–186, 2024
- [2] Ke Wang, Junting Pan, Weikang Shi, Zimu Lu, Houxing Ren, Aojun Zhou, Mingjie Zhan, and Hongsheng Li. Measuring multimodal mathematical reasoning with math-vision dataset. Advances in Neural Information Processing Systems, 37:95095–95169, 2024a.
- [3] Yunzhuo Hao, Jiawei Gu, Huichen Will Wang, Linjie Li, Zhengyuan Yang, Lijuan Wang, and Yu Cheng. Can mllms reason in multimodality? emma: An enhanced multimodal reasoning benchmark. arXiv preprint arXiv:2501.05444, 2025.
- [4] Xiang Yue, Tianyu Zheng, Yuansheng Ni, Yubo Wang, Kai Zhang, Shengbang Tong, Yuxuan Sun, Botao Yu, Ge Zhang, Huan Sun, et al. Mmmu-pro: A more robust multi-discipline multimodal understanding benchmark. arXiv preprint arXiv:2409.02813, 2024b.
- [5] Jiacheng Chen, Tianhao Liang, Sherman Siu, Zhengqing Wang, Kai Wang, Yubo Wang, Yuansheng Ni, Wang Zhu, Ziyan Jiang, Bohan Lyu, et al. Mega-bench: Scaling multimodal evaluation to over real-world tasks. arXiv preprint arXiv:2410.10563, 2024a.

**Ethical Concerns:**

["NO or VERY MINOR ethics concerns only"]

**Final Justification:**

The reviewer answered all my review comments and the questions raised by the rebuttal, and used experiments to prove the effectiveness of their method, so I increased the score accordingly.

**Limitations:**

See in Quesions.

**Quality:**

2

**Strengths And Weaknesses:**

Strengths:
- Novel Structured Reasoning Design: The decomposition of reasoning into steps with Name, Thought, and Reflection components, enforced by special tokens, addresses the lack of structure in existing VLM reasoning chains, improving clarity and alignment with visual information.

- Comprehensive Framework: The integration of SFT, PRM, and iterative DPO provides a holistic solution for enhancing VLM reasoning, with rigorous data construction (ShareGPT-Step-300K) and annotation methods (Monte Carlo estimation, LLM-as-Judge) ensuring robustness.

Weaknesses:
See in Quesions.

---

> ### Author Rebuttal · Authors · 2025-07-28
>
> Dear reviewer gk2j,
>
> Thank you for your feedback and help in improving our work. Below we have responsed to every question individually. In the discussion phase, please enlighten us on anything we can do to further improve the paper.
>
> **Question 1**
>
> We would like to highlight two key differences between our framework and prior reasoning methods in LLMs: **i)** Step-level reasoning and fine-grained rewards. Existing LLM (and VLM) approaches typically generate a complete reasoning chain and evaluate its quality based solely on the correctness of the final answer. For example, in LLM-based DPO, preference pairs are constructed based on final-answer's correctness. In contrast, our approach leverages a PRM to assign scores to intermediate reasoning steps and the final answer, enabling the construction of preference pairs that reflects both intermediate reasoning quality and outcome correctness. This fine-grained supervision provides a more informative and nuanced learning signal. **ii)** Visual information. A key distinction between VLMs and LLMs lies in the incorporation of visual information. In VLMs, the reasoning process can reference concrete visual evidence from the image. To support this, we introduce a *visual reflection* mechanism into the design of reasoning steps, allowing the model to explicitly incorporate and verify visual context during reasoning. This capability is absent in standard LLM settings, where no visual modality is available as an external referent.
>
> **Question 2**
>
> Following your suggestion, we have added the latest Qwen2.5-VL as a baseline as shown in the table below. Although we build from a weaker base model (InternVL-2.5-MPO), our CoS-InternVL still have competitive performances against Qwen2.5-VL. **Despite the competitive results, we would like to emphasize that achieving state-of-the-art performance is not the primary goal of this work.** Instead, our focus is on exploring and validating the effectiveness of fine-grained reasoning and reward mechanisms, and on empirically characterizing their unique properties. Our core contribution lies in establishing a complete and transparent framework that we believe can serve as a foundational tool for the VL community to further investigate fine-grained reasoning chains. And we will release all the components to the public, facilitating future work on step-level reasoning and fine-grained rewards.
>
> |                 | MathVista | MMStar | MMMU | M3CoT | AI2D | ChartQA | Average |
> | --------------- | --------- | ------ | ---- | ----- | ---- | ------- | ------- |
> | LLaVA-OneVision | 63.2      | 61.7   | 48.8 | 52.3  | 81.4 | 80.0    | 64.6    |
> | Qwen2-VL        | 58.2      | 60.7   | 53.7 | 57.8  | 83.0 | 77.4    | 65.1    |
> | Insight-V       | 59.9      | 61.5   | 50.2 | 61.5  | 79.8 | 81.5    | 65.7    |
> | Qwen2.5-VL      | 68.2      | 63.9   | 58.6 | 77.2  | 83.9 | 87.3    | 73.2    |
> | CoS (ours)      | 67.8      | 63.5   | 55.5 | 81.0  | 84.9 | 87.4    | 73.4    |
>
> **Question 3**
>
> Our experiments are conducted using a medium-scale 8B model, and we compare against baselines with similar parameter sizes (7B, 8B, and 11B). We appreciate the suggestion to explore larger-scale models, but training a 72B model is prohibitively expensive for a school lab. For example, even initiating SFT on a 72B model requires at least 32 A100-80G GPUs, not to mention the cost of iterative DPO and extensive empirical studies. More importantly, we would like to **highlight that 8B is a standard setting for developing and validating new techniques and are commonly adopted in the VLM and reasoning literature[1,2,3]**. The insights and conclusion obtained from studying models at this scale are meaningful and generalizable. Thanks for your understanding of the computational constraints.
>
> **Question 4**
>
> The splitting steps are completely determined by the model itself, that is, the model itself generates the reasoning of any number of steps according to the current question. And the name of each step is also not pre-defined manually, but generated by the model itself. This is the original intention of our step based reasoning, we do not restrict the number of steps of reasoning and let the model reason end-to-end. In section 4.3.5 and Figure 6 of the paper, we analyzed the change curves of the steps of different methods during training, and got some interesting findings such as *longer reasoning steps do not necessarily lead to better performance*. Please refer to section 4.3.5 for details of reasoning length.
>
> **Question 5**
>
> Thank you for the suggestion. We have added evaluation results on MMMU-Pro, EMMA, and MegaBench as suggested. Our CoS outperforms basline consistently on these new benchmarks, further demonstrating our design's effectiveness. As for MathVerse and MathVision, we do not report results on these datasets to ensure a fair comparison, as they are already included in our constructed SFT dataset Share-Step-300K.
>
> |                 | MMMU-Pro | EMMA | MegaBench | Average |
> | --------------- | -------- | ---- | --------- | ------- |
> | InternVL2.5-MPO | 34.5     | 22.9 | 30.6      | 29.3    |
> | Cos-InternVL    | 36.5     | 23.5 | 35.8      | 31.9    |
>
> [1] Insight-V: Exploring Long-Chain Visual Reasoning with Multimodal Large Language Models.
> [2] R1-VL: Learning to Reason with Multimodal Large Language Models via Step-wise Group Relative Policy Optimization.
> [3] R1-Onevision: Advancing Generalized Multimodal Reasoning through Cross-Modal Formalization.

---

> > ### Comment · Reviewer_gk2j · 2025-08-01
> > **Some further questions**
> >
> > In Question 2, which Qwen2.5-VL do you choose?
> >
> > And I'd definitely give you bonus points if you could show me that choosing the Qwen2-VL or 2.5-VL as a base model and continuing to use your approach yields consistently positive results.

---

> > > ### Author Response · Authors · 2025-08-03
> > > **Further Response and Experiment**
> > >
> > > Dear reviewer gk2j,
> > >
> > > Thanks for your reply. In our response to Question 2, we use the official Qwen2.5-VL-7B, which is of similar size to our Cos (8B).
> > >
> > > Following your suggestion, we add experiments using **Qwen2.5-VL-7B** as the base model. As shown in the table below, our **Cos-Qwen2.5-VL** has achieved consistent performance improvement on multiple benchmarks, which further demonstrates the effectiveness of our approach.
> > >
> > > |                  | size | MathVista | MMStar | MMMU | M3CoT | AI2D | ChartQA | Average  |
> > > | ---------------- | ---- | --------- | ------ | ---- | ----- | ---- | ------- | -------- |
> > > | LLaVA-OneVision  | 7B   | 63.2      | 61.7   | 48.8 | 52.3  | 81.4 | 80.0    | 64.6     |
> > > | Qwen2-VL         | 7B   | 58.2      | 60.7   | 53.7 | 57.8  | 83.0 | 77.4    | 65.1     |
> > > | Insight-V        | 7B   | 59.9      | 61.5   | 50.2 | 61.5  | 79.8 | 81.5    | 65.7     |
> > > | InternVL-2.5-MPO | 8B   | 65.0      | 60.7   | 53.8 | 67.5  | 84.2 | 85.0    | 69.4     |
> > > | CoS (ours)             | 8B   | 67.8      | 63.5   | 55.5 | 81.0  | 84.9 | 87.4    | 73.4     |
> > > | Qwen2.5-VL       | 7B   | 68.2      | 63.9   | 58.6 | 77.2  | 83.9 | 87.3    | 73.2     |
> > > | **CoS-Qwen2.5-VL (ours)**       | 7B   | 70.4      | 66.3   | 58.9 | 83.7  | 84.9 | 87.9    | **75.3** |

---

### Official Review · Reviewer_S6Gd · 2025-07-03

**Clarity:** 3
**Significance:** 2
**Originality:** 2
**Rating:** 4
**Confidence:** 4

**Summary:**

This paper introduces a straightforward pipeline designed to enhance the general-purpose capabilities of Multimodal Large Language Models (MLLMs). The proposed method involves three main stages: first, synthesizing step-level reasoning data; second, training a Process Reward Model (PRM) on this data; and finally, leveraging the PRM-generated data for training the MLLM using Direct Preference Optimization (DPO).

**Questions:**

Please see the weaknesses. I will adjust my score based on the rebuttal.

**Ethical Concerns:**

["NO or VERY MINOR ethics concerns only"]

**Final Justification:**

Although the innovation of this paper is somewhat limited, it is technically solid and supported by sufficient experiments, which may provide some inspiration to the community. I tend to give it a borderline-accept score.

**Limitations:**

yes

**Quality:**

3

**Strengths And Weaknesses:**

Pros:

- The paper is well-written, presenting a clear and logical progression of ideas, supported by effective illustrations.

- The study is comprehensive, thoroughly addressing key aspects of the pipeline, including data construction and training strategy.

Cons:

- Methodological Novelty: While the empirical contributions are appreciated, the methodological novelty appears to be a primary concern. The pipeline primarily integrates existing techniques such as structured Chain-of-Thought (CoT) data synthesis, PRM training, and DPO. The authors should more explicitly articulate the specific innovations their work introduces beyond the novel application and combination of these established methods. Highlighting the distinction between contribution (the overall pipeline) and innovation (the unique technical advancements) would strengthen the paper.

- Analysis of Model Scaling: The results presented in Table 3 are counterintuitive. It is unclear why the larger InternVL-2.5-38B model does not achieve a higher Answer Accuracy compared to its smaller counterparts.

- Inclusion of Stronger Baselines: The evaluation could be made more robust by including stronger, more relevant baselines. For instance, reinforcement learning methods like PPO or GRPO, when combined with a rule-based verifier/reward model, have demonstrated significant improvements in the reasoning abilities of MLLMs. A comparative analysis against such methods under fair settings would provide a clearer perspective on the performance and advantages of the proposed Iterative DPO approach.

- Significance of Performance Gains: The performance improvement of Iterative DPO over the Supervised Fine-Tuning (SFT) baseline appears marginal. The paper would benefit from a more detailed discussion and analysis of these results to better substantiate the significance and practical benefits of the proposed DPO stage.

---

> ### Author Rebuttal · Authors · 2025-07-28
>
> Dear reviewer S6Gd,
>
> Thank you for your feedback and help in improving our work. Below we have responsed to every question individually. In the discussion phase, please enlighten us on anything we can do to further improve the paper.
>
> **Weakness 1**
>
> **For the technical novelty aspect**, our key innovation lies in the step-level structured design, and the consequent fine-grained RL training and test-time scaling with step-level rewards. Prior attempts on structured CoT reasoning, such as LLaVA-CoT[1], explicitly decompose the reasoning into components like *summary*, *caption*, *reason*, and *conclusion*. However, similar to traditional CoT, these methods rely solely on the correctness of the final answer to evaluate the intermediate reasoning quality, which is clearly suboptimal. In contrast, the core motivation of our work is to enable accurate assessment of intermediate reasoning steps, thereby facilitating more effective RL training and test-time scaling. For example, the DPO used in Insight-V[2] select positive and negative sample pairs based only on the answer, while we choose high-quality reason paths based on PRM scores for every single step and the answer, making the training more effective as shown in the experiments. This step-level design is simple and straightforward, but it serves as the foundation for fine-grained evaluation of reasoning quality and distinguishes our method from prior approaches from the technical perspective.
>
> **For the contribution aspect**, we propose a complete and transparent framework for studying the feasibility, empirical performance, and unique training/inference properties of fine-grained step-level reasoning chains. And we plan to release all components to the community to facilitate future research in fine-grained reasoning. In the revision, we will explicitly highlight the distinctions between technical novelty and contributions to help readers better appreciate the significance of our work.
>
> **Weakness 2**
>
> While the results in Table 3 may appear counterintuitive, they are in fact consistent with the underlying training dynamics.
>
> Notably, **evaluating intermediate reasoning steps is substantially more challenging than evaluating the final answer** for three reasons: **i)** Higher semantic complexity: Step-level evaluation requires a deeper and more holistic understanding of the question. The model must assess whether each step contributes constructively toward the final answer. Similarly, annotating process data is also more cognitively demanding for human annotators[3]. **ii)** More supervision signals. As shown in equation 1 in the paper, PRM is trained to minimize the sum of cross-entropy losses over all intermediate steps and the final answer. A complete reasoning trace contains multiple steps but only a single answer, thus accurately assessing all the intermediate steps is much harder than only assessing the answer. **iii)** Answer memorization. As highlighted in recent studies[4,5], LLMs often exhibit "answer hacking"—producing correct answers via memorized question or patterns rather than genuine understanding. In contrast, intermediate steps exhibit higher variance across instances and are less amenable to memorization.
>
> As we know, **neural networks are inherently short-cut learners**[6]. During training, smaller models (i.e., 8B PRM) tend to prioritize optimizing for final answer accuracy, as it offers a more accessible gradient path to loss reduction. Improving step-level predictions poses a greater optimization challenge. The 38B PRM, with its higher capacity, is better able to model and optimize intermediate reasoning quality. Consequently, it achieves greater improvements in step-level accuracy, which more effectively reduces the total loss—resulting in the performance pattern observed in Table 3.
>
> To verify the consistency of our core finding in this paper, we add an additional experiment using the 8B-PRM. Compared with the  of 38B-PRM, it has a lower accuracy of steps and a higher accuracy of answers. Despite its limited capacity, the results support the same conclusion: jointly optimizing rewards for both intermediate reasoning and final answers yields superior performance.
>
> |                       | MathVista | MMStar | M3CoT | Average |
> | --------------------- | --------- | ------ | ----- | ------- |
> | LLaVA-NeXt-SFT        | 51.4      | 54.7   | 67.4  | 57.8    |
> | Answer (PRM-38B)      | 53.1      | 57.3   | 69.7  | 60.0    |
> | Answer (PRM-8B)       | 53.3      | 57.3   | 70.1  | 60.2    |
> | Outcome               | 53.5      | 58.1   | 70.0  | 60.5    |
> | Step&Answer (PRM-38B) | 54.7      | 58.9   | 71.7  | 61.8    |
> | Step&Answer (PRM-8B)  | 54.6      | 58.3   | 71.5  | 61.5    |
>
> **Weakness 3**
>
> Thanks for your suggestion. The core focus of our work is to investigate the impact of fine-grained rewards on reasoning performance—specifically, comparing *step-level reasoning with fine-grained supervision* against the traditional CoT paradigm with final-answer outcomes. The optimization of the RL algorithm itself is not the primary focus of this work. However, we agree that a more relevant baseline can strengthen this paper, thus we add a new experiment comparing against more relevant baselines as suggested.
>
> Specifically, No Reason denotes a setting where the model performs direct question answering without any explicit reasoning; Direct Prompt means directly prompting VLMs to reason step by step (as in [1]); Cos corresponds to our proposed method using step-level structured reasoning with fine-grained rewards. All methods are trained using the same SFT and RL datasets to ensure a fair comparison. For No Reason, we use only the question-answer pairs (without intermediate reasoning thought) from the ShareGPT-Step-300K dataset for SFT. For Direct Prompt, we concatenate all steps of reasoning (i.e., "thoughts") from each instance into a single reasoning sequence. Since both No Reason and Direct Prompt can not output strict step-level thought, we cannot assess their intermediate reasoning quality, so we apply the outcome reward (i.e., correctness of the final answer) during RL training. We use LLaVA-NeXt as the base model. There are two key observations: i) Models trained with reasoning-augmented SFT data (i.e., Direct Prompt and Cos) clearly outperform models trained with QA-only data (No Reason) on reasoning benchmarks; ii) Our Cos yields much greater improvements during RL compared to outcome-only reward settings.
>
> |                   | Reason | RL Reward | MathVista | MMStar | M3CoT | Average     |
> | ----------------- | ------ | --------- | --------- | ------ | ----- | ----------- |
> | LLaVA-NeXt        | -      | -         | 45.9      | 43.1   | 45.6  | 44.9        |
> | No Reason SFT     | ×      | -         | 49.1      | 53.9   | 62.1  | 55.0        |
> | No Reason RL      | ×      | outcome   | 51.5      | 56.4   | 63.4  | 57.1 (+2.1) |
> | Direct Prompt SFT | √      | -         | 51.2      | 54.9   | 66.3  | 57.5        |
> | Direct Prompt RL  | √      | outcome   | 53.1      | 58.2   | 69.3  | 60.2 (+2.7) |
> | CoS SFT           | √      | -         | 51.4      | 54.7   | 67.4  | 57.8        |
> | CoS RL            | √      | PRM       | 54.7      | 58.9   | 71.7  | 61.8 (+4.0) |
>
> These additional results further support our central motivation: the step-level reasoning paradigm, combined with fine-grained rewards, consistently outperforms traditional CoT approaches that rely solely on final-answer supervision. Regarding RL algorithm, we agree that including experiments with GRPO or PPO would further strengthen the work. However, both GRPO and PPO require online deployment of the PRM, which is much more computationally expensive. In contrast, DPO offers a more resource-efficient alternative by enabling offline preference modeling using PRM. We plan to add a set of GRPO experiment in the final version to further validate our findings. Thanks for your understanding of the computational and time constraints.
>
> **Weakness 4**
>
> We would like to clarify that our intention is not to demonstrate that iterative DPO is necessarily better than SFT, actually they are both effective in enhancing VLM's reasoning abilities. We conduct experiments on two base models, the weaker LLaVA-NeXt and the stronger InternVL2.5-MPO. For LLaVA-NeXt, both SFT and iterative DPO lead to performance gains and SFT gains are better than iterative DPO gains (+8.7 v.s. +3.4), indicating that for relatively weaker VLMs, SFT is more effective and direct in enhancing the model's performance. In contrast, for the stronger InternVL2.5-MPO, the performance improvement from SFT alone is relatively limited (+1.6). However, after iterative DPO, the model can still achieve a notable performance boost (+2.4), reaching state-of-the-art. This indicates that for stronger VLMs, the benefits of further improvement through SFT are marginal, while RL is more essential and effective for enhancing strong model’s reasoning capabilities. **From the perspective of significance, improving over a strong baseline is more difficult and meaningful than improving over a weak baseline.** For the strong baseline, iterative DPO's gains exceed SFT's gains (+2.4 v.s. +1.6), demonstrating the effectiveness of fine-grained reward RL in improving the upper limit of reasoning ability.
>
> [1] LLaVA-CoT: Let Vision Language Models Reason Step-by-Step.
> [2] Insight-V: Exploring Long-Chain Visual Reasoning with Multimodal Large Language Models.
> [3] Let's Verify Step by Step.
> [4] SoK: Memorization in General-Purpose Large Language Models.
> [5] Rethinking Memorization in LLMs: On Learning by Rote vs. with Understanding.
> [6] Shortcut Learning in Deep Neural Networks.

---

> > ### Comment · Reviewer_S6Gd · 2025-08-05
> >
> > Thank you for the comprehensive rebuttal. Most of my concerns have been addressed. However, I still find the technical novelty of the work to be somewhat limited. Additionally, the absence of GRPO/PPO experimental results weakens the empirical validation of the approach. Therefore, I will maintain my current score for now. If the authors could provide these missing results, I would be open to slightly raising my score accordingly.

---

> > > ### Author Response · Authors · 2025-08-08
> > > **GRPO/PPO experiments**
> > >
> > > Dear Reviewer S6Gd,
> > >
> > > Thanks for your reply! Here are our clarification and GRPO/PPO experiments.
> > >
> > > For technical novelty, we understand that the judgment of novelty is from person to person and **would like to thank you for appreciating our empirical analysis and contribution, which is exactly our central focus and core contribution**. We do believe that our dataset, framework and extensive empirical studies can make positive contributions to the VLM reasoning community.
> > >
> > > Following your suggestion, we add GRPO/PPO experiments in the table below. We deploy the 8B PRM as our reward-server due to the time constraint. For GRPO experiment settings, we set batchsize to 128, rollout.n to 5, lr to 1e-5 and run 100 steps starting from the SFT model. For PPO experiment settings, we set batchsize to 128, lr to 1e-5 and run 100 steps starting from the SFT model.
> > >
> > > |                | Reward      | MathVista | MMStar | M3CoT | Average         |
> > > | -------------- | ----------- | --------- | ------ | ----- | --------------- |
> > > | LLaVA-NeXt-SFT | N/A         | 51.4      | 54.7   | 67.4  | 57.8            |
> > > | outcome PPO    | outcome     | 53.1      | 57.9   | 70.7  | 60.6            |
> > > | CoS (PRM PPO)  | step&answer | 54.1      | 58.6   | 71.5  | **61.4 (+0.8)** |
> > > | outcome GRPO   | outcome     | 54.3      | 57.9   | 71.4  | 61.2            |
> > > | Cos (PRM GRPO) | step&answer | 56.1      | 59.2   | 73.5  | **62.9 (+1.7)** |
> > >
> > > As shown above, our fine-grained reward RL outperforms outcome reward RL consistently using both GRPO and PPO, further strengthening our conclusions. In our final version, we will also add the larger 38B PRM's GRPO/PPO results, which are expected to have further advantages and merits.
> > >
> > > Great thanks for your engagement and suggestions to improve our paper, and we sincerely hope the above experiments can eliminate your remaining concern and raise to a positive rating. If there is anything further we can do or clarify, please let us know.

---

> > > > ### Comment · Reviewer_S6Gd · 2025-08-08
> > > >
> > > > Thank you for providing the additional results, which make the work more comprehensive. These results, together with the experimental details, should be incorporated into the revised version. I have accordingly raised my score.

---

> ### Comment · Area_Chair_uewq · 2025-08-05
> **Please respond to the authors' rebuttal**
>
> Hi,
> As the deadline of author-reviewer discussion is approaching, could you please check the authors' rebuttal and respond to it accordingly?
>
> Thanks
> AC

---

> ### Author Response · Authors · 2025-08-08
> **Further Response**
>
> Dear Reviewer S6Gd,
>
> Great thanks for your time and suggestions! We will revise the paper as suggested in the final version.

---

### Decision · Program_Chairs · 2025-09-17

**Decision:**

Accept (poster)

**Comment:**

Although one reviewer raised concerns about the presentation quality, all reviewers are consistently positive about the paper’s solid technical contributions and strong empirical results. Therefore, I recommend acceptance.